# Asynchronous Heart Rate Variability Biofeedback Protocol Effects on Adolescent Athletes’ Cognitive Appraisals and Recovery-Stress States

**DOI:** 10.3390/jfmk8030094

**Published:** 2023-06-30

**Authors:** Philippe Vacher, Quentin Merlin, Guillaume Levillain, Laurent Mourot, Guillaume Martinent, Michel Nicolas

**Affiliations:** 1Research Center for Education Learning and Didactics (EA 3875), Faculty of Sports Science, University Brest, 29200 Brest, France; philippe.vacher@univ-brest.fr (P.V.); guillaume.levillain@univ-brest.fr (G.L.); 2Laboratory Psy-DREPI (EA 7458), University Bourgogne Franche-Comté, 21000 Dijon, France; quentin_merlin@etu.u-bourgogne.fr (Q.M.); michel.nicolas@u-bourgogne.fr (M.N.); 3EA 3920-Prognostic Markers and Regulatory Factors of Cardiac and Vascular Diseases and Plateforme EPSI, Université de Franche-Comté, 25000 Besançon, France; laurent.mourot@univ-fcomte.fr; 4Laboratory L-VIS (EA 7428), University of Claude Bernard Lyon 1, 69200 Lyon, France

**Keywords:** adolescent athletes, heart rate variability, biofeedback, recovery, stress

## Abstract

This study examined the effect of an asynchronous heart rate variability biofeedback (HRV-BFBasync) protocol on national-level adolescent swimmers’ cognitive appraisals and recovery-stress states during a six-week ecological training period. A polynomial mixed-effects multilevel regression analysis approach was used with 27 adolescent national-level swimmers randomly assigned to an intervention group (n = 14) and a control group (n = 13). Six waves of assessments of cognitive appraisals and recovery-stress states were completed during six weeks of training preparation in ecological conditions. The results revealed that the HRV-BFBasync protocol significantly predicts lower levels of biopsychosocial stress states and cognitive stress. However, no significant effects were found for biopsychosocial recovery scales and cognitive perceived control. The results suggested that total stress states, sport-specific stress, and cognitive perceived stress evolutions are a function of polynomial time third-degree interactions with HRV-BFB protocol. Overall, this study suggested that the HRV-BFBasync protocol leads adolescent athletes to experience lower biopsychosocial and cognitive stress levels during training periodization. Our results also suggest that HRV-BFB induces complex evolutions over time for stress and recovery states but does not have a predictive function for the recovery states and perceived control.

## 1. Introduction

Adolescent elite athletes are exposed to various stressors (e.g., sports-specific, scholarly, social), making stress and recovery management a precious resource to prevent maladaptive fatigue and injury and increase training and competitive training performance. From a biopsychosocial perspective [1], stress and recovery are viewed as multidimensional and interactive [2]. In this field, stress is defined as a destabilization/deviation from the norm of biological/psychological systems (i.e., psychophysical balance) [2]. This unspecific reaction is accompanied by a series of symptoms, including psychological (e.g., emotional disturbances), physiological, and behavioural changes. Recovery is defined as a process that restores the abilities related to performance. Kellmann and Kallus (2001) describe recovery as characterized by intra- and inter-individual components, such as psychological, physiological and social processes. Today, recovery is regarded as a multifaceted (e.g., psychological, physiological) restorative process relative to time [3]. Based on it, stressors will destabilize one’s biopsychosocial states, leading to fatigue and increased tiredness. A recent consensus statement on recovery in sports underlines that recovery strategies can be split into physiological and psychological [3]. Physiological technics mainly refer to regeneration in sports and exercise contexts (e.g., cold water immersion, nutrition diet, sleep). These regenerative techniques usually follow exercise stressors (i.e., training or competition) which have induced physical fatigue. On a psychological level, several strategies have been investigated, such as coping strategies or psychological relaxation techniques, to reduce mental fatigue and prevent under-recovery in athletes. To date, investigating applied technics that help athletes cope with their stress states and optimize their recovery ability is a critical target for athletes and researchers. More particularly, it seems interesting to investigate how the body’s psychological and physiological parts may interact to optimize individual stress response and their stress-recovery balance.

In recent years, brain–heart interactions developed in the allostatic stress model [4] suggest that psychological arousal and autonomic regulation over the cardiovascular system influence one another. These works provide knowledge on the stress process, leading to the stress-recovery balance described above. In this work, the body–brain pathway includes physiological systems’ feedback and feed-forward processes that integrate bodily information. These systems, such as the autonomic nervous system (ANS) and the hypothalamic–pituitary–adrenal axis (HPA), promote adaptation—called allostasis by Sterling and Eyer, 1988 [5].

In this approach, individual differences are due to two main factors. The first is how one perceives and interprets the situation. This cognitive interpretation of the situation is close to the cognitive appraisals introduced in the transactional model of stress and coping [6]. Cognitive appraisal of a situation is a core psychological process explaining within-person adaptation variations. Primary appraisal addresses whether the situation is personally relevant and leads to perceived stress. The primary appraisal is immediately mirrored by a secondary appraisal, which evaluates an individual’s internal and external resources for managing a situation. The secondary appraisal leads to the perceived control (PC) of stressful situations. According to the model of Lazarus and Folkman (1984), PC can be defined as the degree to which one believes they can determine their behaviour and internal states, influence their environment, and/or produce desired outcomes [7]. The second aspect of individual differences in response to stressful situations is the body’s condition. This aspect covers the physical condition, metabolic balance, genetic predisposition, developmental stage, and gender.

Still based on a biopsychosocial perspective and considering the allostatic model, several processes are involved in the stress response. Brain–heart interactions consensually refer to the central autonomic network (CAN) [8,9]. The CAN connects the central autonomic system to the periphery bidirectionally via the immune system, the HPA axis and the ANS [10]. Researchers have investigated how to use the bidirectional characteristics of these relationships to optimize individuals’ stress response to environmental challenges. Specifically, the ANS can be differentiated into a sympathetic branch (which mobilizes the so-called “flight or fight” response to a stressor) and a parasympathetic branch (which restores individual resources). The literature underlines that the two branches do not work in opposition but in a complementary manner leading to the fact that the ANS can only perform properly if both parts are working in a well-regulated way [11]. Furthermore, the sympathetic and parasympathetic branches can be active simultaneously: when parasympathetic activity increases, sympathetic activity does not automatically decrease [12,13,14,15]. This nervous activity can be indexed non-invasively using an electrocardiogram and calculating heart rate interbeat variability, named heart rate variability (HRV) [16,17]. To summarize, reduced HRV indicates relatively high sympathetic activation, disturbed regulation of the ANS, and inadequate adaptation of the cardiovascular system [18]. It is also a sign of chronic stress, the depletion of energy reserves, and autonomic imbalance [13]. Reversely, a relatively high HRV reflects an adaptive organism, optimal energy reserves, and a good state of autonomic control mechanisms [12,13]. Considering the dynamic systems approach, which focuses on increasing individuals’ self-regulatory capacity by inducing a physiological shift that is reflected in the heart’s rhythms, McCraty and Childre (2010) theorize that rhythmic activity in living systems reflects the regulation of interconnected biological, social, and environmental networks [19]. They suggested that HRV encodes information communicated across multiple systems, which helps synchronize the system [13]. The coherence model suggests that the amount of HRV that is mediated by efferent vagal fibres reflects self-regulatory capacity. In this approach, the afferent pathways from the heart and cardiovascular system are central due to the significant degree of afferent cardiovascular input to the brain and the consistent generation of dynamic patterns generated by the heart [13]. It has been shown that important inter-individual differences exist in baroreflex frequency, which can vary from 0.07 Hz to 0.11 Hz [20], corresponding to 4.5 to 6.5 breath cycles per minute [21]. This supports the development of the resonance frequency approach [20,22,23]. These inter-individual differences also lead to the need for biofeedback, especially to HRV biofeedback protocols (HRV-BFB). The applied protocols are based on the works of Lehrer [20,24], who described the resonant frequency breathing technique. They demonstrated that biofeedback training is useful for increasing the amplitude of respiratory sinus arrhythmia (RSA). Biofeedback maximally increases the amplitude of heart rate oscillations only at approximately 0.1 Hz. Cardiac coherence/resonance yields greater reflex efficiency, afferent brain stimulation, and an optimized psychophysiological state [25,26,27]. These protocols have been largely conducted in patients suffering from various pathologies (such as chronic pain or psychological diseases). In this context, the literature has repeatedly shown positive effects for patients suffering from chronic physiological [28] and psychological diseases and stress disorders [29]. HRV-BFB also improves anxiety, depression, anger and athletic/artistic performance [30,31]. In the sport context, it has been shown that the HRV-BFB protocol based on resonance frequency may help to improve cardiac variability during the recovery period in short-term effort recovery and improve the perception of recovery and the perception of physical exertion [32]. Similarly, it has been shown that biofeedback protocols (EEG vs. HRV) lead to stress reduction; however, the nature of the protocol used in this study does not allow an understanding of the specific value of HRV-BFB [33]. Complementary recent works have shown that presleep HRV-BFB improved some measures of autonomic function, mood, and sleep quality in Chinese Olympic bobsleigh athletes [34]. In contrast, another study showed that HRV-BFB did not improve coherence, psychological, or performance variables [35]. Finally, the necessity of applied protocols led researchers to develop shorter HRV-BFB protocols (5 min of practice twice a day rather than 40 min daily as generally recommended in the seminal literature), showing positive effects on the autonomic function but no effects on the psychological states [36].

Athletes are predisposed to cope regularly with stress and are exposed to pressure time and schedules that constitute an obstacle to developing the practice of HRV-BFB in their seminal length. Considering the pressured schedules of athletes and the necessity to develop functional applied protocols for athletes to help them cope with their strenuous environment, the present study was designed to examine the effects of six weeks of HRV-BFB_async_ training on the potential benefits of cognitive appraisal and biopsychosocial recovery-stress states of adolescent elite athletes. HRV-BFB_async_ is developed on the same rationale as regular resonant HRV-BFB, but the identification of the 0.1 Hz pic was identified and trained as a posteriori of the monitoring. We hypothesize that the intervention group will report lower cognitive and biopsychosocial stress during the protocol, unlike the control group.

## 2. Materials and Methods

### 2.1. Population

Twenty-seven adolescent national-level swimmers were randomly assigned to an experimental training group (n = 14, females = 6, males = 8; Mage = 15.2 ± 2.48 yrs; Mheight = 171.13 ± 9.44 cm; Mweight = 60.7 ± 13.7 kg; Mcompetition experience = 8.53 ± 1.88 yrs) and a no-treatment control group (n = 13, females = 5, males = 8; Mage = 15.7 ± 2.15 yrs; Mheight = 168.34 ± 9.86 cm; Mweight = 59.4 ± 11.1 kg; Mcompetition experience = 6.36 ± 1.75 yrs) before further information was given about the study. They were members of the same global training group and were exposed to the same sport/scholar context and environmental/structural stressors. They trained 24 h per week (specific training = 20 h; physical preparation = 4 h). We chose to focus on elite adolescent swimmers because the literature repeatedly shows that this population is characterized by high training load volume and intensity that lead to stress and recovery imbalance [37,38,39], making these athletes an at-risk population in terms of biopsychosocial state imbalance. Regarding the literature, we choose not to record the menstrual cycles. Indeed, even though it is well recognized that menstrual cycles may influence HRV measurement, recent research has shown no relationships in the HRV-BFB protocol [36].

A priori power analysis was performed using Power IN Two-level designs software designed to estimate standard errors of regression coefficients in two-level hierarchical linear models for power calculations [40]. If α is chosen at 0.05, we expect a medium effect size, and if a power of 0.80 is desired, then a sample of 25 participants along six measurement points is required for the model, including most of the variables in the present study.

The study was part of the ASDP project validated by the ethics committee of Alliance Universitaire Bretagne under the number 2303077 and was carried out following the Declaration of Helsinki. After comprehensive verbal and written explanations of the study, all the subjects gave their written informed consent to participate. For minors, parents/guardians gave full written informed consent.

### 2.2. Materials

#### 2.2.1. Biopsychosocial Recovery-Stress States

The short French version of the Recovery-Stress Questionnaire for Athletes (RESTQ-36-R-Sport) was used to measure the recovery-stress state of the athletes [41,42]. We used the general, specific and total scores of stress and recovery in order to adopt a holistic perspective of the athletes’ recovery and stress states. The response scale asked participants to rate the frequency of each item over the preceding three days/nights on a scale of 0 (never) to 6 (always). Cronbach’s alphas ranged from 0.60 to 0.94 for the different times’ measurements; 36 items (see Table 1).

#### 2.2.2. Cognitive Appraisals—Perceived Stress and Control

An adaptation to the sporting context of the mastery scale (M.S.) [43] was used to assess the level of PC. and an adaptation of the perceived stress scale (PSS) [44]. As per the central tenets of stress theory [45], the items within the M.S. and the PSS were relatively independent of content specific to any particular situation and population [43]. For a short and quick administration, we used the six French items validated by Martinent and Nicolas (2017) and reworded “competition” into “training” to refer to the training context and not to the competition context (e.g., “I feel able to cope with the stress of the training”; “I have the resources to cope with training pressures”; “I feel able to master the challenges that I could meet during training”) [46]. The scores indicated the extent to which the athletes agreed with these statements on a 6-point Likert scale (1 = strongly disagree to 6 = strongly agree). Cronbach’s alphas ranged from 0.74 to 0.94 for the different times’ measurements; 6 items) (see Table 1).

### 2.3. Experimental Design

HRV-BFB protocol was assigned to the experimental group (HRV-BFB group) for six weeks. The second author of this study, a mental coach, delivered this intervention. Quickly, session 1 (introduction) aimed to introduce fundamental concepts (i.e., stress, recovery, respiratory sinus arrhythmia, coherence and resonant breathing), develop athletes’ adhesion and illustrate the effects of breathing at 0.1 Hz. Session 2 (skill development) aimed to help athletes to choose the appropriate breathing rhythm and to introduce the smartphone application and polar H7 monitor and technics parts of the protocol. Session 3 (skill development) aimed to underline the psychophysiological differences between the “natural breathing phase” and the “coherence breathing phase” and to identify individual resonant breathing rhythms. The last part of this session was dedicated to planning with athletes 2 × 10 min per day (using Breath/Breath + smartphone app (10 min in the morning upon awakening + 10 min in the evening just before bedtime)). Sessions 4 and 5 were performed mid-week. The goal was to debrief and control the conformity of the protocol realization. A resonant breath monitoring session was planned at each session (i.e., 5 min natural breathing phase; resonant exploration phase: two minutes at 6.5, 6, 5.5, 5, 4.5 breaths/min; 5 min natural breathing phase). The goal was to follow up on the resonant breath rhythm for the next week. Finally, session 6 was conducted before the end of the protocol. It consists of a debriefing of the protocol after one week of autonomy (without regulation or discussion with the mental coach). During the protocol, the no-treatment control group continued their regular training program without the experimental training. Please refer to Table 2 for a complete description of the protocol and the content of the sessions.

### 2.4. Procedure

#### Data Analysis

Multilevel growth curve analyses (MGCA) examined the linear and/or quadratic and/cubic trajectories of athletes’ recovery-stress states and cognitive appraisals [47]. All analyses were conducted using the R package labelled lme4 [48]. Separate analyses were conducted for each of the psychological (i.e., general, specific and total scores of stress and recovery; perceived control and stress) states. Multilevel models extend multiple regressions to nested data (hierarchically structured data). Specifically, repeated measurements (Level 1 units of analysis) were nested within individuals (Level 2 units of analysis). Multilevel models are a flexible approach that can be applied to evaluate inter-individual differences in intra-individual changes over time (i.e., each participant has their own curve). Thus, by taking into account the hierarchical structure of the data, multilevel models provide unbiased estimates of the parameters [47].

Firstly, a series of two-level models estimated the average growth and the individual differences in growth. At Level 1, time (linear trajectory, quadratic and cubic) was entered as a predictor to estimate the average intercept (β0), the average linear growth (β1), the average quadratic growth (β2) and the average cubic growth (β3). The intercept should be interpreted as the level of the state of the athletes at the start of the training periodization. The random effects of the intercept and linear slope were included in each model. Because training load is recognized as the strongest structural stressor on the athletes, the subjective training load was introduced as a predictor for the model (sRPE) (model 1: simple linear time effect; model 2: linear, quadratic and cubic effects of time and sRPE) [49,50]. Secondly, the present study also sought to test the role of intervention in the time trajectory of recovery-stress states and cognitive appraisals. Thus, we included the interactions of time (of the dependent variables’ trajectories) group based on the rationale that a significant interaction indicates that the psychological state trajectory is significantly different across the experimental versus the control groups (and thus an effect of the intervention on the psychological states) (model 3: linear, quadratic and cubic effects of time + sRPE + interaction effects with the group for the time effects; see Table 3).

## 3. Results

Interaction Effect of Time with Group

The results have shown a significant simple effect of the HRV-BFBasync protocol on the specific (β = −15.96) and the total scores of stress states (β = −13.09). Similarly, a simple negative effect of the HRV-BFBasync was found for perceived stress (β = −16.35).

The statistical analysis has shown significant interaction effects of time with HRV-BFBasync for the specific (marginal R² = 0.42) and the total scores of stress states (marginal R² = 0.14). Similarly, a significant interaction effect of time with HRV-BFB was found for perceived stress (β = 7.62; marginal R² = 0.20). Considering this result, statistical analysis showed that while perceived stress continuously decreased after the start of the protocol, the control group showed an increase since times 2 and 5 (i.e., end of the skill development period and home-work period for the intervention group; see Figure 1). Please refer to Table 3 for the complete presentation of the results.

## 4. Discussion

This study aimed to adapt a resonant protocol based on HRV-BFB to the training context, characterized for many training centres by limits on their 1) financial support to buy biofeedback materials, and 2) expertise in autonomic nervous system data treatment and exploitation. Due to the promise of the resonant works and their potential benefits for athletes [20,22,51], we proposed in this seminal work to develop an asynchronous protocol in which the resonant pick is monitored using the protocol of Vaschillo et al., 2006 [21]. Furthermore, considering that many training centres are fitted with HRV-monitoring tools such as the Polar H7 (or similar tools), we proposed using such tools in developing our protocol.

Then, this study aimed to investigate the effects of a 6-week HRV-BFBasync protocol divided into three key periods: weeks 1 to 3 were dedicated to psychoeducation and skill learning about (1) cardiac coherence and resonance, (2) breathing technics and (3) the identification of individual breathing rates in order to obtain 0.1 Hz resonant frequency. Weeks 4 and 5 were dedicated to the life course implementation of HRV-BFBasync (2 × 10 min per day). Finally, week 6 was a learning week with full autonomy (see Table 2 for a complete protocol presentation). As hypothesized, we observed significant effects of HRV-BFBasync on adolescent athletes’ biopsychosocial stress states (i.e., sport-specific and global stress states) and cognitive stress. Interestingly, no effects were found for the biopsychosocial recovery states and the cognitive perception of control over the training process. These results should be closely interpreted in light of the protocol’s design (mainly asynchronous modality, 2 × 10 min per day, six weeks). Indeed, as underlined in recent years, the design of HRV-BFB training seems critical in obtaining obvious benefits [36].

Based on the CAN, HRV-BFBasync seems to have effects similar to those previously observed in the literature on stress with a reduction in the global stress states for the intervention group [52]. In our study, this effect is observed for the cognitive and biopsychosocial components of stress. These results align with the fact that in the CAN, the autonomic nervous control supposes an independent, automatic self-governing system responsible for maintaining body physiology through low-level, occasionally orchestrated patterns of reflex responses. Furthermore, the autonomic control supports the complexity of organism-level behaviours necessary for survival (e.g., interoceptive signalling allowing motivational states that lead to motivational states, themselves associated with pleasant/unpleasant affective valence). This engenders specific autonomic and behavioural responses, which reflect the prioritization, selection, and execution of allostatic adaptive strategies [5,53]. This way, the autonomic nervous control is integrated with affective, motivational and cognitive processes [54]. Then, it is fair to argue that the HRV-BFBasync activated the resonance mechanisms that reduce the cognitive appraisal of stress and the states of stress [22]. This is particularly interesting for adolescent athletes exposed to a wide range of stressors in the sport context leading to cognitive and emotional perturbations [38] along the training process. Complementarily, it has been shown that intense psychophysiological solicitations induce the dissociation of the HPA/SAM coordination [55,56], which is associated with training load and parasympathetic markers in HRV in the adolescent athlete population [57]. Therefore, using HRV-BFBasync seems to be of interest to counter at least some of the adverse effects induced by intensive training and may help athletes better cope with the training process by preserving cognitive and biopsychosocial resources.

The second significant result of our study is that no effects were found for the recovery part of the biopsychosocial states, as well as for the second cognitive appraisal (PC). To our knowledge, we found only one study investigating how an HRV-BFB protocol may influence recovery states in a sports context [32]. Despite this study showing a positive impact on recovery states, relying on these results is not viable because the HRV-BFB training design and context are very different from ours. Indeed, while in the study of Perez-Gaido et al., 2021, the purpose was to investigate the impact of HRV-BFB on short-term effort recovery (showing an improvement in the psychophysiological adaptation after intense aerobic exercise provided by the HRV-BFB), our protocol is designed for a more chronic aspect of the training process. In our design, no significant effects were found, suggesting that in an actual life implementation, HRV-BFBasync does not increase the athletes’ perception of recovery and their cognitive perception of internal and/or external resources. This result aligns with the recent pilot study by Weber et al., 2022, which examined the effects of a 4-week biofeedback intervention on collegiate student athletes [35]. Although this study does not investigate athletes’ recovery states from a biopsychosocial perspective, they showed no effects on the recovery of sleep quality, insomnia and chronotype approach. In this study, no effects were found on recovery, suggesting that short HRV-BFB does not lead to modification in the psychophysiological resources of athletes.

## 5. Conclusions

In conclusion, it can be suggested that integrating HRV-BFBasync leads to modifying the cognitive and behavioural stress part in charge of the CAN [54]. Using such protocols may help athletes to better cope with the stress component of their training program (i.e., specific parts of the stress states component) and how they cognitively interpret the potential of the stress of the training program. In this way, it seems interesting to combine such techniques, which will optimize the stress component of the psychophysiological states, with other techniques that may boost the recovery component. Further studies are needed before concluding on the effects of such programs. However, HRV-BFBasync seems to be an up-and-coming tool to answer the need for feasibly applied protocols to support athletes and coaches in their performance research.

Despite the original approach of this study, it has some limitations. First, our results are specific to the adolescent intensive training population and cannot be extended to the general population. Second, even though we based our protocol on the seminal works of Vaschillo and current HRV-BFB protocols, the asynchronous design may open discussion on the fact that this protocol is effectively a biofeedback protocol. We chose to stay in the biofeedback perspective because the breathing pace was based on the same process used in conventional resonant frequency protocols, leading to an individualization of the 0.1 Hz frequency. Third, we focused our measurements on questionnaires to target variables of interest and respond to the population’s organizational constraints. It would be interesting to use mixed methods (qualitative and quantitative) to obtain the athletes’ feelings about implementing such protocols to optimize their integration into daily life. Fourth, our protocol focuses on adolescent swimmers only, as this population is very representative of adolescent athletes who have to meet high-volume training loads, but future research must investigate such protocols on other sports and levels before generalizing our results. Finally, this protocol remains exploratory because we have not found any other study using asynchronous biofeedback. Thus, future studies need to confirm that there are effects only on the stress part of the model and not only on the recovery part.

## Figures and Tables

**Figure 1 jfmk-08-00094-f001:**
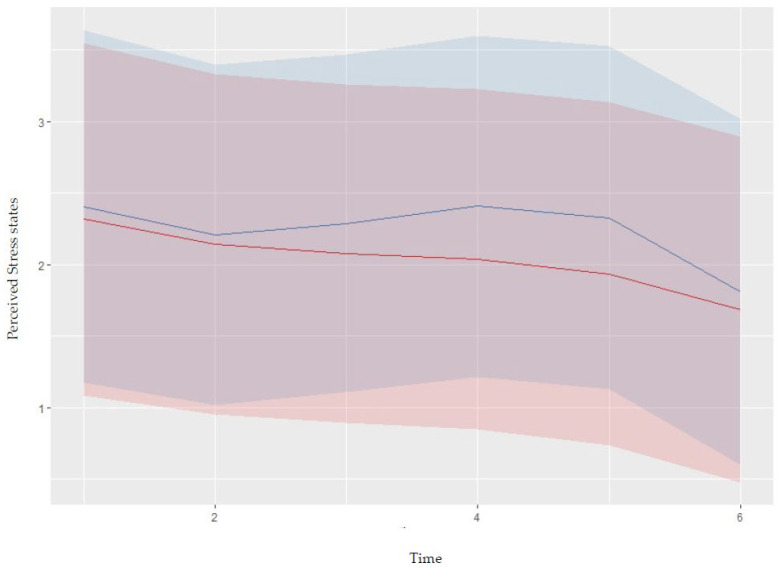
Graphic illustration of the interaction effects of HRV-BFB with time for the first cognitive appraisal (i.e., perceived stress states). Red curves: intervention group; blue curves: control group.

**Table 1 jfmk-08-00094-t001:** Descriptive statistics for the six measurement waves.

		Time 1		Time 2		Time 3		Time 4		Time 5		Time 6	
	Group	M (SD)	α	M (SD)	α	M (SD)	α	M (SD)	α	M (SD)	α	M (SD)	α
sRPE	I	4782 (1447)	−	3857 (1277)	−	3174 (1380)	−	3684 (1056)	−	3429 (860)	−	1708 (895)	−
	C	4345 (969)		3316 (1402)		3410 (1080)		3426 (1245)		3306 (1111)		1809 (732)	
General stress	I	2.07 (1.12)	0.90	2.26 (1.09)	0.89	1.87 (0.96)	0.92	2.03 (0.82)	0.90	1.81 (0.82)	0.92	1.60 (0.78)	0.94
	C	2.76 (0.99)		2.66 (1.17)		2.63 (0.93)		2.38 (0.58)		2.31 (0.86)		1.84 (0.95)	
General recovery	I	3.72 (1.17)	0.89	3.70 (1.33)	0.93	3.58 (1.02)	0.90	3.76 (0.69)	0.82	3.93 (0.65)	0.87	3.87 (0.97)	0.87
	C	2.94 (1.05)		3.08 (1.08)		3.15 (1.21)		3.56 (1.22)		3.68 (1.13)		3.41 (1.18)	
Sport-specific stress	I	1.74 (0.78)	0.60	1.69 (0.92)	0.84	1.50 (0.70)	0.90	1.77 (0.78)	0.91	1.50 (0.71)	0.91	1.44 (0.75)	0.92
	C	3.05 (1.86)		2.29 (1.09)		2.34 (0.77)		1.94 (0.48)		1.95 (0.47)		1.81 (0.76)	
Sport-specific recovery	I	3.32 (1.06)	0.90	3.32 (1.06)	0.90	3.45 (1.12)	0.80	3.50 (1.08)	0.74	3.29 (0.90)	0.79	3.51 (3.04)	0.77
	C	2.63 (1.17)		2.56 (0.95)		2.72 (1.07)		3.33 (1.32)		2.91 (1.42)		3.04 (1.24)	
Total stress	I	1.90 (0.92)	0.83	1.98 (0.95)	0.92	1.69 (0.75)	0.87	1.90 (0.66)	0.82	1.66 (0.64)	0.82	1.52 (0.69)	0.88
	C	2.90 (1.37)		2.47 (1.02)		2.49 (0.78)		2.16 (0.48)		2.13 (0.63)		1.83 (0.82)	
Total recovery	I	3.52 (1.09)	0.94	3.51 (1.15)	0.94	3.52 (1.02)	0.88	3.63 (0.74)	0.78	3.61 (0.68)	0.82	3.69 (0.92)	0.80
	C	2.79 (1.05)		2.82 (0.90)		2.94 (1.00)		3.44 (1.10)		3.30 (1.19)		3.22 (1.14)	
Perceived Control	I	4.21 (1.24)	0.85	4.59 (1.17)	0.91	4.59 (1.16)	0.89	4.69 (1.05)	0.90	4.67 (1.02)	0.88	4.53 (0.95)	0.94
	C	4.12 (1.17)		4.00 (1.06)		4.11 (1.13)		4.57 (1.05)		4.44 (1.37)		4.61 (1.09)	
Perceived Stress	I	2.29 (1.47	0.88	2.44 (1.31)	0.89	1.90 (1.31)	0.91	1.89 (1.10)	0.81	1.86 (1.09)	0.82	1.56 (0.70)	0.74
	C	2.38 (1.22)		2.18 (1.31)		2.42 (1.11)		2.23 (0.93)		2.15 (0.78)		1.70 (0.66)	

Note. Time 1 to Time 6: measurement points of the protocol; sRPE: subjective internal training load; M—mean; SD—standard deviation; α—Cronbach’s alpha for internal consistency of questionnaires.

**Table 2 jfmk-08-00094-t002:** The HRV-BFB asynchronous protocol for applications with adolescent athletes.

Weeks	Session Goals	Tools	Modality
Week 1 (introduction)	Session 1 (60 min) Create adhesion to the intervention. Help athletes to understand the notion of recovery and stress. Introduction of the recovery principles (e.g., active, passive, proactive component, multicomponent aspects). Introduction to resonant breathing specificities and the “respiratory sinus arrhythmia”.	Individual sessions based on question-and-answer games centred on athletes’ stress, recovery and breathing. Video supports and self-participation to explore breathing rhythms and amplitude. Screenshot of the HRV modification due to coherence and resonance.	Classroom/ Group
Week 2 (skill development)	Session 2 (60 min) Help athletes to choose the appropriate breathing rhythm. Introduction to the coherence technic and advantage for stress and recovery (6 cycles per minute). Team discussion on the effects of the coherence effect, feelings and somatic perceptions. Introduction to the method. Monitoring HRV using polar H7 at six cycles per minute for 10 min using Breathe/Breathe + smartphone application to obtain an initial estimate of resonant frequency (5 min natural breathing phase; 10 min coherence breathing phase; 5 min natural breathing phase).	Screenshot/iconography. Polar H7. Kubios HRV standard. Breathe/Breathe+ smartphone application. Gym mats.	Classroom/ Group
Week 3 (skill development)	Session 3 (60 min) Debriefing on the session two monitoring. Mental coaches identify on the screen the effects of coherent breathing. Mental coaches underline the differences between the “natural breathing phase” and the “coherence breathing phase”. The mental coach points out the frequency peak at 0.1 Hz if it exists. After that, using polar H7, identification of the “resonant frequency”: (5 min natural breathing phase; resonant exploration phase: two minutes at 6.5, 6, 5.5, 5, 4.5 breaths/minute; 5 min natural breathing phase). The mental coach realizes HRV analysis using Kubios HRV and informs the athlete of their resonant frequency (i.e., the maximum amplitude of RSA is achieved). The mental coach asks athletes to practice slow breathing at their resonant frequency twice daily using the Breath/Breath + smartphone app (10 min in the morning upon awakening + 10 min in the evening just before bedtime) during the two-next weeks.	Screenshot/iconography. Polar H7. Kubios HRV standard. Breathe/Breathe + smartphone application. Gym mats.	Individual
Week 4–5 (home practice)	Session 4 and 5 (2 × 10 min per day + 40 min face-to-face): One time per week (in mid-week), an individual debriefing is conducted with the mental coach to control the conformity of the protocol. During this 40 min session, a new resonant frequency mapping is conducted (i.e., 5 min natural breathing phase; resonant exploration phase: two minutes at 6.5, 6, 5.5, 5, 4.5 breaths/minute; 5 min natural breathing phase). If necessary, the individual breath pace for resonant frequency is adjusted for the next of the protocol.	Screenshot/iconography. Polar H7. Kubios HRV standard. Breathe/Breathe + smartphone application. Gym mats.	Individual
Week 6. (Autonomy)	Session 6 (60 min): Debriefing on feelings, difficulties and advantages of the technique. Sharing of experiences. The mental coach informs the athletes that a final measurement will be taken within a month. Afterwards, athletes are free to practice depending on the benefits and limitations experienced during the last weeks.	Screenshot/iconography. Gym mats.	Classroom

**Table 3 jfmk-08-00094-t003:** Unstandardized parameters estimates of the growth curve model 3.

	Training Load	Stress-Recovery Balance						Cognitive Appraisals	
	sRPE	GS	SS	TS	GR	SR	TR	PC	P.S.
		Fixed effects—Estimates (Standard errors)							
Intercept	6919.17 (9579.99)	6.01 (4.91)	10.95 (4.80) *	8.23 (3.87)	0.27 (5.30)	−3.44 (5.14)	−1.96 (4.55)	−0.89 (6.04)	10.42 * (6.61)
sRPE	−	0.00 (0.00)	−0.00 (0.00)	0.00 (0.00)	0.00 (0.00)	0.00 * (0.00)	0.00 (0.00)	0.00 (0.00)	0.00 (0.00)
Time _[1st degree]_	−3538.63 (19,975.33)	−7.04 (0.11)	−15.57 (9.98)	−10.83 (8.04)	5.52 (10.01)	12.49 (10.69)	9.44 (9.46)	10.56 (12.52)	−16.13 (9.55)
Time _[2nd degree]_	1382.18 (14,608.34)	5.26 (7.45)	10.68 (7.29)	7.68 (5.887)	−3.91 (8.06)	−9.98 (7.82)	−7.17 (6.92)	−8.22 (9.14)	11.40 (6.98)
Time _[3rd degree]_	−301.96 (4835.00)	−1.81 (2.46)	−3.44 (2.41)	−2.55 (1.94)	1.25 (2.67)	3.60 (2.59)	2.48 (2.29)	2.87 (3.02)	−3.68 (2.31)
Group	−13,565.79 (130,393.40)	−11.54 (6.72)	−15.96 * (6.57)	−13.09 * (5.28)	0.77 (7.27)	6.87 (7.05)	4.65 (6.22)	2.45 (8.25)	−16.35 * (6.31)
Time _[1st degree]_ * Group	29,061.14 (27,304.04)	22.81 (14.00)	30.41 * (13.69)	25.36 * (11.01)	−0.35 (15.15)	−14.06 (14.70)	−8.72 (12.96)	−6.29 (17.17)	33.53 * (13.12)
Time _[2nd degree]_ * Group	−21,117.54 (19,963.30)	−16.53 (10.23)	−21.92 * (10.00)	−18.36 * (8.04)	0.33 (11.07)	11.05 (10.75)	6.66 (9.47)	5.83 (12.55)	−23.84 * (9.59)
Time _[3rd degree]_ * Group	6873.37 (6602.85)	5.43 (3.38)	7.23 * (3.31)	60.67 * (26.60)	−0.16 (3.66)	−3.85 (3.56)	−2.28 (3.13)	−2.24 (4.15)	7.62 * (3.17)
		Random effects—Variance (Standard deviation)							
σ^2^	1,091,471 (1044.74)	0.28 (0.53)	0.26 (0.51)	0.17 (0.41)	0.33 (0.57)	0.31 (0.56)	0.24 (0.49)	0.42 (0.64)	0.24 (0.49)
τ_00subjects_	485,676 (696.90)	0.41 (0.64)	0.39 (0.63)	0.23 (0.48)	0.44 (0.67)	0.38 (0.61)	0.85 (0.92)	0.49 (0.70)	1.68 (1.29)
τ_11subjects.time_	9763 (98.81)	0.03 (0.16)	0.03 (0.16)	0.02 (0.14)	0.03 (0.18)	0.02 (0.01)	0.00 (0.04)	0.04 (0.20)	0.02 (0.15)
τ_11subjects.sRPE_	−	0.00 (0.00)	0.00 (0.00)	0.00 (0.00)	0.00 (0.00)	0.00 (0.00)	0.00 (0.00)	0.00 (0.00)	0.00 (0.00)
		Performance Model							
Marginal R²	0.412	0.315	0.42	0.14	0.19	0.28	0.31	0.13	0.20
logLik	−1190.0	−151.83	−146.85 **	−123.69 *	161.60	−155.16	279.82	−176.39	−153.93 *

Notes. S.E. = standard errors; S.D. = standard deviations; *β*_0*j*_ is the average level of psychological states for individuals; γ_00_ = intercept of level-2 regression predicting *β*_0*j*_; γ_10_ = intercept of level-2 regression predicting *β*_1*j*_; σ^2^ = var(r_ij_) variance in level-1 residual (i.e., variance in r_ij_); τ_00_ = var(U_0j_) variance in level-2 residual (i.e., variance in U_0j_). GS: general stress; SS: sport-specific stress; TS: total stress; GR: general recovery; SR.: sport-specific recovery; TR.: total recovery; PC: perceived control; PS: perceived stress. * *p* < 0.05; ** *p* < 0.01.

## Data Availability

Data available on request due to restrictions eg privacy or ethical. The data presented in this study are available on request from the corresponding author. The data are not publicly available because the request was not specified to the ethics committee when the project was submitted.

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
