# Peer review of "Asynchronous Heart Rate Variability Biofeedback Protocol Effects on Adolescent Athletes’ Cognitive Appraisals and Recovery-Stress States"

_jfmk, 2023, doi:10.3390/jfmk8030094_

Round 1

Reviewer 1 Report

Overall, this is a well-designed research project with an interesting topic. The authors provided thorough introduction and background.

Here are some comments and suggestions

1)      The introduction seems relatively long and some of the information was not directly related to the present study. I suggest the authors summarize different theories/models of current studies. On the other hand, the introduction can be strengthened by building stronger links between the background information and the purpose of this present study.

2)      Line 181-183: the authors cited 2 references to support that menstrual cycles are independent of outcome measurements. However, these 2 references also support that menstrual cycles can influence heart rate variability (HRV), which is also a major outcome measurement of the present study. Please clarify the reason that menstrual cycles and oral contraceptive pills information were not considered in females in this study.

3)      Table 1 (line 204): please add footnote for Time 1 to6, respectively and sRPE. The context spelled out sRPE later, but it should be spelled out and explained for Table 1 since it appeared first in the manuscript.

4)      Line 210: what is P.C.?

5)      Line 222: what was the protocol for the control group? Did the control group receive any information during the 6 weeks?

6)      Line 228: what was the smartphone application that being used by the participants?

7)      The results section reported several tables; however, the tables were not very explanatory themselves and the context was not clear about the findings either. Please consider modifying the results section by explaining the findings in a way that readers can follow and understand the differences in outcomes between the 2 groups.

8)      Instead of reporting all the numbers from the methodological data in the results section, please summarize the findings by comparing outcomes between groups.

9)      Figure 1 (Line 310): the figures were not very clear since there were 2 colored lines and 3 shades on each figure. The context related to figure 1 did not explain much about the findings, either. Please elaborate the findings related to figure 1.

Minor editing of English language required.

Author Response

Overall, this is a well-designed research project with an interesting topic. The authors provided thorough introduction and background.

Dear reviewer, thank you for your time revising our study and for your feedback quality and accuracy. We have done our best to respond to your comments and incorporate your requested corrections. We sincerely hope our corrections will allow the article to be accepted.

Here are some comments and suggestions

  • The introduction seems relatively long and some of the information was not directly related to the present study. I suggest the authors summarize different theories/models of current studies. On the other hand, the introduction can be strengthened by building stronger links between the background information and the purpose of this present study.

Thank you for your suggestion. We deleted some parts of the introduction to reduce the introduction length .

However, activating these systems can also cause problems for the body and the mind if they are over- or under-active [4]. The price of allostasis is the consumption of resources (i.e., psychological and physiological), which can promote pathophysiology, named allo-static load [4]. An interesting point in the allostatic model is that there are significant dif-ferences in how individuals respond to potentially stressful situations.

L67-71. New section:  These systems, such as the autonomic nervous system (ANS) and hypothalamic-pituitary-adrenal axis (HPA), promote adaptation − called allostasis by Sterling & Eyer, 1988 [5]. In this approach, individual differences are due to two main factors. The first is how one's perceives and interprets the situation.

L.84-89. Still based on a biopsychosocial perspective and considering the allostatic model, several processes are involved in stress response. Brain-heart interactions consensually refer to the central autonomic network (CAN) [8,9]. The CAN involves inter-connected telencephalon, diencephalon, and brainstem areas that control the sympathetic outputs via preganglionic neurons in the brainstem vagal nuclei and the spinal cord [8,10]. The CAN primarily processed the moment-to-moment integration of the information collected from central and peripheral systems [11]. Firstly, areas of the CAN integrate bodily sensations with emotional and goal-related autonomic responses (i.e., insular and anterior cingulate cortex; the limbic system including the amygdala). Secondly, the CAN controls complex endocrine, autonomic, and sleep responses through the hypothalamus. Thirdly, the CAN play a crucial role in pain modulation and behavioural responses to environmental demands through the periaqueductal grey [11]. Then, it has been shown that the CAN reflects one's capability to adapt to environmental challenges [12].

Because tThe CAN connects the central autonomic system to the periphery bidirectionally via the immune, the HPA axis and the ANS [13]. Researchers have investigated how to use the bidirectional characteristics of these relationships to optimize individuals' stress response to environmental challenges. Specifically, the ANS can be differentiated into a sympathetic branch (which mobilizes the so-called "flight or fight" response to a stressor) and a parasympathetic branch (which restores individual resources).

Line 110-115. The coherence model suggests that the amount of HRV that is mediated by efferent vagal fibres reflects self-regulatory capacity. In this approach, the afferent pathways from the heart and cardiovascular system are central due to the significant degree of afferent cardiovascular input to the brain and the consistent generation of dynamic patterns generated by the heart [13]. It has been shown that important inter-individual differences exist in baroreflex frequency, which can vary from 0.07Hz to 0.11Hz [20], corresponding to 4.5 to 6.5 breath cycles per minute [21].

They defined the term "coherent heart rhythm as a relatively harmonic (sine-wave-like) signal with a very narrow, high-amplitude peak in the Low Frequency (L.F.) region of the HRV power spectrum with no major peaks in the Very Low (VLF) or High Frequencies (H.F.) regions. Coherence is assessed by identifying the maximum peak in the 0.04–0.26 Hz range of the HRV power spectrum, calculating the integral in a window 0.030 Hz wide, centred on the highest peak in that region, and then calculating the total power of the entire spectrum. The coherence ratio is formulated as (Peak Power/[Total Power – Peak Power])" [23]. It has been shown that important inter-individual differences exist in baroreflex frequency, which can vary from 0.07Hz to 0.11Hz [24], corresponding to 4.5 to 6.5 breath cycles per minute [25].

  • Line 181-183: the authors cited 2 references to support that menstrual cycles are independent of outcome measurements. However, these 2 references also support that menstrual cycles can influence heart rate variability (HRV), which is also a major outcome measurement of the present study. Please clarify the reason that menstrual cycles and oral contraceptive pills information were not considered in females in this study.

Indeed, it is well known that menstrual cycles influence HRV. However, to the best of our knowledge, no specific study has investigated how menstrual cycles may impact resonance frequency protocol results. We chose not to consider menstrual cycles and oral contraceptive pills information regarding the results of a recent study realized by Deschodt-Arsac et al. (2018), which showed little if any, disturbances linked to menstrual cycle.

Line 162-168. To clarify, we suggest modifying the sentences: " We chose to focus on elite adolescent swimmers because literature repeatedly shows that this population is characterized by high training load volume and intensity that lead to stress and recovery imbalance [37–39], making these athletes an at-risk population in terms of biopsychosocial state imbalance. Regarding the literature, we choose not to record the menstrual cycles. Indeed, even though it is well recognized that menstrual cycles may influence HRV measurement, recent research has shown no relationships in HRV BFB protocol [36].

 Because the recorded information suggests little, if any, disturbances linked to menstrual cycles, and because our study mainly focused on subjective states, we do not report the menstrual cycles and oral contraceptive pills of the females [41,42].

  • Table 1 (line 204): please add footnote for Time 1 to6, respectively and sRPE. The context spelled out sRPE later, but it should be spelled out and explained for Table 1 since it appeared first in the manuscript.

I am not sure to understand the matter with time 1 to 6. We suggest the following modification of the table note.

Note. Time 1 to Time 6: measurement points of the protocol; sRPE : subjective internal training load ; M, mean; S.D., standard deviation; α, Cronbach's alpha for internal consistency of questionnaires.

  • Line 210: what is P.C.?

Line 196. We apologize for the error. It is PC (i.e., perceived control). We delete the "."

Thank you.

  • Line 222: what was the protocol for the control group? Did the control group receive any information during the 6 weeks?

Thank you for your question. As we say it in the manuscript, the two groups are parts of the same training group: "They were members of the same global training group and were exposed to the same sport/scholar context and environmental/structural stressors." Then, they follow the same training and scholarship protocol than the interventional group. However, in terms of psychological intervention. To be clear on this point, we add Line 156 "and a no-treatment control group (n=13, females = 5, males = 8; Mage = 15.7±2.15yrs; Mheight = 168.34±9.86cm; Mweight = 59.4±11.1kg; Mcompetition experience = 6.36±1.75yrs) before further information was given about the study.

  • Line 228: what was the smartphone application that being used by the participants?

Line 217-219. This information is included in the experimental design section (section 2.3): "The last part of this session was dedicated to planning with athletes the 2*10min per day (using Breath / Breath+ smartphone app (10 minutes in the morning upon awak-ening + 10 minutes in the evening just before bedtime). "

  • The results section reported several tables; however, the tables were not very explanatory themselves and the context was not clear about the findings either. Please consider modifying the results section by explaining the findings in a way that readers can follow and understand the differences in outcomes between the 2 groups.
  • Instead of reporting all the numbers from the methodological data in the results section, please summarize the findings by comparing outcomes between groups.

We understand that questions 7 and 8 are in the same line, and we propose to rewrite in a synthesized manner the results section. We focused the results section on the HRV-BFB results and deleted additional information about the null model and the simple effect of time. Because of the multilevel characteristics of the statistical analysis we made, Table 3 can not be proposed simply… We have to give the fixed and random effects and the model's characteristics. In the same line, we focus the figure on the more straightforward pattern we obtained and delete the two other graphics to make him more likeable. We sincerely hope that our result section will be more understandable by deleting additional content.

9)      Figure 1 (Line 310): the figures were not very clear since there were 2 colored lines and 3 shades on each figure. The context related to figure 1 did not explain much about the findings, either. Please elaborate the findings related to figure 1.

 We suggest focusing the figure on the most typical pattern in our results. Please see Figure 1. The results section is now:

The statistical analysis has shown significant interaction effects of time with HRV-BFBasync for the specific (marginal R² = 0.42) and the total scores of stress states (marginal R² = 0.14). Similarly, a significant interaction effect of time with HRV-BFB was found for perceived stress (β = 7.62; marginal R² = 0.20). Considering this result, statistical analysis showed that while perceived stress continuously decreased after the start of the protocol, the control group showed an increase since times 2 and 5 (i.e., end of the skill development period and home-work period for the intervention group; see Figure 1). Please refer to Table 3 for a complete results presentation.

Reviewer 2 Report

A well-designed and written manuscript that investigated and summarized heart rate variability in athletes during cognitive appraisals and recovery stress state. But I want to raise few questions here.

      1.     Add graphical abstract in the beginning.

2.     Why are study group or athletes selected only from national level swimmers? Why not from other sports domain like athletics, tennis …etc.? How come (HRV-BFBasync) protocol have a similar predictive function for the recovery states and perceived control for every other sports? Justify?

3.     Authors chose unequal sample sizes (Male and females) in both control and experimental groups to design this study. Justify your design.

4.     How did you handle your outliers in this study? It looks like there is no outliers. How come its possible?

5.     Disturbances linked to menstrual cycles and oral contraceptive pills of the females were not reported as per authors this study focused on subjective states. Please explain or add a reference which supports your claim that menstrual cycles and oral contraceptive pills of the females does not affect subjective states of mind.

6.      Entire manuscript filled with lot of tables, try to convert some tables into graphs.

7.     Authors found no significant effects for the recovery part of the biopsychosocial states, as well as for the second cognitive appraisal. This one because as these results are specific to the adolescent intensive training population. Justify? 

           Overall, the explanation of the proposed hypothesis of this work is very loud and clear.

Author Response

A well-designed and written manuscript that investigated and summarized heart rate variability in athletes during cognitive appraisals and recovery stress state. But I want to raise few questions here.

  1. Add graphical abstract in the beginning.

Line 35. A graphical abstract is now proposed. We hope it meets your expectations.

  1. Why are study group or athletes selected only from national level swimmers? Why not from other sports domain like athletics, tennis …etc.? How come (HRV-BFBasync) protocol have a similar predictive function for the recovery states and perceived control for every other sports? Justify?

The athletes were chosen in a sport with high training volumes and intensity. The population is exposed to high training loads, which have been linked in the literature to stress and recovery states destabilization. This population is, therefore, at risk from the point of view of overtraining and internal psychological and physiological balance. We have targeted the national level because, given the age of our population, these are athletes who are already practising very intensively (24 hours a week). Based on this volume and level, we can consider these athletes elite in the term's broadest sense (access to the top level). As for the fact that we do not have any other population, let us underline the difficulty of accessing top-level athletes for interventional protocols. Of course, it is an interesting prospect for future studies.

Line 162-165. We suggest adding in the population section: "We chose to focus on elite adolescent swimmers because literature repeatedly shows that this population is characterized by high training load volume and intensity that lead to stress and recovery imbalance [37–39], making these athletes an at-risk population in terms of biopsychosocial state imbalance."

  1. Collette, R.; Kellmann, M.; Ferrauti, A.; Meyer, T.; Pfeiffer, M. Relation between Training Load and Recovery-Stress State in High-Performance Swimming. Front. Physiol. 2018, 9, 1–14, doi:10.3389/fphys.2018.00845.
  2. Vacher, P.; Nicolas, M.; Martinent, G.; Mourot, L. Changes of Swimmers’ Emotional States during the Preparation of National Championship: Do Recovery-Stress States Matter? Front. Psychol. 2017, 8, 1–11, doi:10.3389/fpsyg.2017.01043.
  3. Vacher, P.; Martinent, G.; Mourot, L.; Nicolas, M. Elite Swimmers’ Internal Markers Trajectories in Ecological Training Conditions. Scand. J. Med. Sci. Sports 2018, 28, 1866–1877, doi:10.1111/sms.13200.

To avoid overinterpreting our results, we add at the end of the paper: " Third, we focused our measurements on questionnaires to target variables of interest and respond to the population's organizational constraints. It would be interesting to use mixed methods (qualitative and quantitative) to obtain the athletes' feelings about implementing such protocols to optimize their integration into daily life. Fourth, our protocol focuses on adolescent swimmers only, this population is very representative of adolescent athletes who have to meet with high volume training loads, but future research must investigate such protocols on other sports and levels before generalizing our results. "

We hope all these precisions will help you better understand our methodology and population choice.

  1. Authors chose unequal sample sizes (Male and females) in both control and experimental groups to design this study. Justify your design.

This protocol is based on a real population in ecological conditions. Then we do not “choose” the population. We do with the population which agrees to participate to the protocole in real conditions / not in laboratory or in a more experimental condition. The training group was composed of 27 athletes. We do not have a choice about this. Then, we randomly attribute males and females to experimental and control groups. The structuration is eight males in each group, six females in the experimental group vs 5 in the control group. The two groups are the more equal we can.

  1. How did you handle your outliers in this study? It looks like there is no outliers. How come its possible?

The fact that there are no outliers is not all that surprising. It has been shown that the relatively small sample used in the present study could, in fact, represent a stringent test of our hypotheses because smaller samples have a higher likelihood of not yielding significant results although the effect exists (see Schweizer G, Furley P. Reproducible research in sport and exercise psychology: the role of sample sizes. Psychol Sport Exerc. 2016;23:114‐122.). Furthermore, this population is very representative (homogeneous) with typical populations that have a high level of practice (practised around 20 hours per week) and expertise (competing at a national level for at least 4 years before the study). All these reasons can explain the consistency of our results.

  1. Disturbances linked to menstrual cycles and oral contraceptive pills of the females were not reported as per authors this study focused on subjective states. Please explain or add a reference which supports your claim that menstrual cycles and oral contraceptive pills of the females does not affect subjective states of mind.

We suggest the following modification of this part of the methodology. It seems that we didn't formulate our thoughts properly. We suggest the following modification:

We chose to focus on elite adolescent swimmers because literature repeatedly shows that this population is characterized by high training load volume and intensity that lead to stress and recovery imbalance [37–39], making these athletes an at-risk popula-tion in terms of biopsychosocial state imbalance. Regarding the literature, we choose not to record the menstrual cycles. Indeed, even though it is well recognized that men-strual cycles may influence HRV measurement, recent research has shown no rela-tionships in HRV BFB protocol [36].

  1. Entire manuscript filled with lot of tables, try to convert some tables into graphs.

We understand that the number of tables may dicourage the reader. Following your comments (you and the other reviewer), we have deleted tables and keep only the last one. In the same line, we focus the figure on the main result. We hope it will answer to your comment.

  1. Authors found no significant effects for the recovery part of the biopsychosocial states, as well as for the second cognitive appraisal. This one because as these results are specific to the adolescent intensive training population. Justify?

This is a very interesting and pertinent question. Unfortunately, we don't have the answer. As we explained in the introduction, few studies have been carried out on recovery with biofeedback protocols. They were initially built around stress mechanisms, which makes our results consistent. On the other hand, the fact that no effect was observed on recovery would tend to indicate an effect on resource preservation rather than resource regeneration. This is rather at odds with the commercial potential of these techniques. In another study currently being submitted, we have observed the opposite effect for a coherence protocol (i.e. without biofeedback). We do not believe that these aspects are specifically linked to our population, but this remains to be demonstrated. We suggest adding the following sentence at the end of the article:

Finally, this protocol remains exploratory because we have not found any other study using asynchronous biofeedback. Thus, future studies need to confirm that there are effects only on the stress part of the model and not only on the recovery part.

Overall, the explanation of the proposed hypothesis of this work is very loud and clear.

Round 2

Reviewer 1 Report

Thank the authors for revising the manuscript.

1. Please spell out "PC" (line 196) since it seems to be the first time that PC appears in the manuscript.

2. Section 2.3 (Line 207) - please add a sentence to clarify that the non-training group continued their regular training session without the experimental training. Since the term "training" were both used to describe these athlete's training and also the experimental training from the present study, a clarification is needed here for the readers.

None.

Author Response

Thank the authors for revising the manuscript.

Once again, thank you for your valuable expertise on our manuscript. In this second round of revision, we acknowledge all your recommendations. Sincerly.

  1. Please spell out "PC" (line 196) since it seems to be the first time that PC appears in the manuscript.

Sorry for this error. Following your remark, we add the bottom modifications :

Line77 – 80. “The secondary appraisal leads to perceived control (PC) of stressful situations. According to the model of Lazarus and Folkman (1984), PC can be defined as the degree to which one believes he can determine his behaviour and internal states, influence his environment, and/or produce desired outcomes [7].”

Line 280 / Notes part (delete points). “GS: General Stress; SS: Sport Specific Stress; TS: Stotal Stress; GR: General Recovery; SR.: Sport Specific Recovery; TR.: Total Recovery; PC: Perceived Control; PS: Perceived Stress. *P < .05; **P < .01; ***P < .001.”

Line 329. “as well as for the second cognitive appraisal (PC).”

  1. Section 2.3 (Line 207) - please add a sentence to clarify that the non-training group continued their regular training session without the experimental training. Since the term "training" were both used to describe these athlete's training and also the experimental training from the present study, a clarification is needed here for the readers.

Line 226-228. “During the protocol, the no-treatment control group continued their regular training program without the experimental training.”